# Comparative Virucidal Activities of Essential Oils and Alcohol-Based Solutions against Enveloped Virus Surrogates: In Vitro and In Silico Analyses

**DOI:** 10.3390/molecules28104156

**Published:** 2023-05-18

**Authors:** Valentina Parra-Acevedo, Raquel E. Ocazionez, Elena E. Stashenko, Lina Silva-Trujillo, Paola Rondón-Villarreal

**Affiliations:** 1Centro de Cromatografía y Espectrometría de Masas—CROM-MASS, Universidad Industrial de Santander, Bucaramanga 680002, Colombia; valentina2218470@correo.uis.edu.co (V.P.-A.); elena@tucan.uis.edu.co (E.E.S.); lina2198192@correo.uis.edu.co (L.S.-T.); 2Facultad de Ciencias Médicas y de la Salud, Instituto de Investigación Masira, Universidad de Santander, Bucaramanga 680003, Colombia; diseno.molecular@udes.edu.co

**Keywords:** virucidal activity, essential oils, disinfectant, enveloped viruses

## Abstract

The large-scale use of alcohol (OH)-based disinfectants to control pathogenic viruses is of great concern because of their side effects on humans and harmful impact on the environment. There is an urgent need to develop safe and environmentally friendly disinfectants. Essential oils (EOs) are generally recognized as safe (GRAS) by the FDA, and many exhibit strong antiviral efficacy against pathogenic human enveloped viruses. The present study investigated the virucidal disinfectant activity of solutions containing EO and OH against DENV-2 and CHIKV, which were used as surrogate viruses for human pathogenic enveloped viruses. The quantitative suspension test was used. A solution containing 12% EO + 10% OH reduced > 4.0 log10 TCID_50_ (100% reduction) of both viruses within 1 min of exposure. In addition, solutions containing 12% EO and 3% EO without OH reduced > 4.0 log10 TCID_50_ of both viruses after 10 min and 30 min of exposure, respectively. The binding affinities of 42 EO compounds and viral envelope proteins were investigated through docking analyses. Sesquiterpene showed the highest binding affinities (from −6.7 to −8.0 kcal/mol) with DENV-2 E and CHIKV E1-E2-E3 proteins. The data provide a first step toward defining the potential of EOs as disinfectants.

## 1. Introduction

Enveloped RNA viruses such as coronavirus, influenza A (HIN1) virus, and Ebola virus are responsible for pandemics and epidemics, which are transmitted primarily through close person-to-person contact as well as through aerosolized respiratory droplets [1,2]. A susceptible person can also be infected by indirect transmission by self-inoculation through the mucous membranes of the nose and mouth by touching contaminated surfaces. Viruses can persist for hours or even days on inanimate surfaces [1]. Therefore, the use of disinfecting agents for surface cleaning and personal care is one of the first-line strategies to limit virus transmission during an epidemic [3].

The World Health Organization recommends alcohol (OH)-based hand sanitizers to control the transmission of human pathogenic enveloped viruses [4]. Generally, OH-based virucidal disinfectants contain high concentrations of ethanol (80% *v*/*v*) or isopropanol (70% *v*/*v*) or a combination of these [5,6]. Because of their lipophilicity, OHs damage the phospholipid membrane of viruses by the delipidation and denaturation of proteins. Although OH-based disinfectants exhibit strong virucidal activity, they have limitations and their excessive use can be a threat to living beings [7,8,9]. OHs are flammable liquids, and prolonged exposure to ethanol causes skin and eye irritation; alcohol evaporates rapidly when exposed to air, thereby reducing the efficacy of the disinfectant; and fomites with prolonged exposure to OH may compromise their integrity. Mitigation strategies are required to reduce these effects.

Essential oils (EOs) distilled from aromatic plants are complex mixtures of monoterpene and sesquiterpene hydrocarbons and oxygenated compounds such as phenols, alcohols, aldehydes, ethers, and ketones [10]. EOs were proposed as starting points for drug discovery to prevent and treat viral infections. This is because numerous EOs exhibit in vitro antiviral activity against pathogenic human enveloped viruses such as herpesvirus, flavivirus, coronavirus, influenza A virus, and human immunodeficiency virus [11,12,13].

EOs have applications in industries other than pharmaceuticals, including the cleaning products industry [10,14,15,16]. EO-based disinfectants have been proposed as sanitizing agents for disinfection [5,14,16]. They can be used as an ingredient in OH-based disinfectants against viruses, reducing the adverse effects of the OH, but maintaining the virucidal action [17,18]. EO and ethanol mixtures were effective in reducing the concentration of viral particles when applied to ceramic, stainless steel, and laminate surfaces [17]. The EO of tea tree (*Melaleuca alternifolia*) combined with ethanol was effective in inactivating feline coronavirus (FCoVII) and the human coronavirus HCoV-OC4 [18].

In the current study, we investigated the virucidal disinfectant activity of an EO blend combined or not with OH against dengue virus type 2 (DENV-2) and chikungunya virus (CHIKV), which were used as pathogenic enveloped RNA virus surrogates. In addition, using an in silico approach, the possible activity of 42 EO compounds against viral envelope proteins was also investigated.

## 2. Results

### 2.1. Test Solutions

Table 1 shows the solutions tested for virucidal disinfectant activity. A single EO blend was used, which contained pure EOs from seven Colombian aromatic plants. In addition, an OH preparation was used, which contained ethanol (70%) and a mixture (25%) of isopropanol and glycerol. Eight solutions containing EO (3%, 6%, and 12%) combined or not with the OH preparation (1%, 5% and 10%) were analyzed. The cytotoxicity assay revealed that none of the test solutions were cytotoxic to Vero cells (Table 2). The cell viability ranged from 80% up to 100%, relative to untreated cells, after incubation of the cells with the lowest dilution (1:10) of each solution.

### 2.2. Virucidal Disinfectant Activity of the Test Solutions

The quantitative suspension test was used to evaluate the antiviral disinfectant activity following the German DVV/RKI guideline [19], and limits were 3.8–5 ± 1.2 and 5.3 ± 0.55 TCID_50_ (log10) per mL of DENV-2 and CHIKV, respectively (Table 3). A reduction factor of 4-log10 was the cutoff value for disinfectant activity [19]. First, we evaluated the activity of solutions containing 12% EO combined or not with OH within 1 min of exposure (Figure 1A). The solution containing 12% EO without OH (12EO) was sufficient to achieve a 3.9-log10 (81.6%) reduction in the DENV-2 titer, but was insufficient to reduce the CHIKV titer. The addition of 1% (12EO + 1OH solution) and 5% (12EO + 5OH solution) OH increased the reduction of DENV-2 to >4-log10 (100%), whereas the addition of 10% OH (12EO + 10OH solution) was required to achieve a 4-log10 (100%) reduction of CHIKV. Next, we evaluated solutions containing EO at concentrations lower than 12% combined with 10% OH after 1 min of exposure (Figure 1B). A reduction of 4-log10 of DENV-2, but not CHIKV, was achieved with the 6% EO + 10% OH (6EO + 10OH) solution, whereas the reduction of both viruses was not observed with the 3% EO + 10% OH (3EO − 10OH) solution. Solutions containing 3% EO without OH (3EO) and 10% OH without EO (10OH) did not show a virucidal effect against either virus after 1 min of exposure.

As the solutions containing 12% EO and 3% EO without OH did not show disinfectant effects against DENV-2 and CHIKV after 1 min of exposure, we assessed the activity of these solutions by increasing the exposure time in four intervals (Table 4). For the 12% EO solution (12EO), a reduction of >4-log10 (100%) of DENV-2 was achieved after 5 min of exposure, whereas a 100% reduction of CHIKV was achieved after 10 min of exposure. For the 3% EO solution, an exposure time of 30 min was required to achieve a 100% reduction in DENV-2 and CHIKV.

### 2.3. Chemical Composition of the EO Blend

Table 5 presents the linear retention indices and relative amounts of compounds in order of their elution on the DB-5MS column. A chromatogram of the EO blend is presented (Appendix A). Forty-two compounds were identified. Monoterpene alcohols (52%) and aldehydes (23.8%) were the most abundant terpenes, especially geraniol (35.4%), citronellal (22.6%), and citronellol (14.1%), followed by monoterpene acetates (7.2%) and hydrocarbons (4.2%). Sesquiterpenoids were identified in low concentration (8.4%) and sesquiterpene hydrocarbons (5.7%), mostly germacrene D and δ-cadinene, were in higher concentration than oxygenated sesquiterpenes.

### 2.4. Molecular Interactions of EO Compounds and Viral Proteins

The DENV particle has a capsid surrounded by a lipid envelope, which contains the envelope (E) and membrane (prM/M) proteins [24]. The forty-two compounds identified in the EO blend were subjected to molecular docking simulation against E and prM/M of DENV-2. Appendix A presents the AutoDock Vina binding energies. Twenty-five (60%) compounds bound to the E protein, of which twelve bound with a strong binding energy (−7.03 to −8.61 kcal/mol) and thirteen with a weak binding energy (−6.0 to −6.7 kcal/mol). The compounds were accommodated in a consensus site corresponding to the detergent beta-octylglucoside (βOG) pocket in the hinge region of the E protein, which formed hydrophobic bonds with amino acid residues (Figure 2). Sesquiterpene hydrocarbons such as cadinene (δ and γ), α-guaiene and α-bulnesene, and the oxygenated sesquiterpene *epi*-α-muurolol showed the strongest binding affinities (−8.10 to −8.61 kcal/mol) with E, followed by monoterpene hydrocarbons (limonene, *p*-cymene, and γ-terpinene: −7.16 to −7.22 kcal/mol) and phenolic compounds (carvacrol and thymol: −7.03 to −7.32 kcal/mol). Other oxygenated sesquiterpenes (farnesol, α-cadinol, and α-eudesmol) and oxygenated monoterpenes (isopulegol, citronellyl acetate, neral, geranial, and citronellol) showed weak (−6.50 to −6.70 kcal/mol) binding affinities with the E protein. Farnesol and α-cadinol formed hydrogen bonds, the first with Gyl190, Leu191, and His282, and the second with Thr48 and Tyr137. Twenty compounds that bind to E were also identified in a previous study on the anti-DENV activity of EOs from other Colombian plant species [25]. Table 6 presents the EO compounds that docked DENV-2 E identified in this study. Docking analyses did not predict the binding of EO compounds to the prM/M protein (docking scores ranged from −4.08 to −5.47 kcal/mol).

The CHIKV particle has a capsid surrounded by a lipid envelope, which contains the E1-E2-E3 glycoprotein complex [26]. The forty-two compounds identified in the EO blend were subjected to molecular docking against E1-E2-E3. Appendix A presents the AutoDock Vina binding energies. Eight of the ten oxygenated sesquiterpenes identified in the EO blend bound to E1-E2-E3, and α-cadinol, α-eudesmol, and caryophyllene oxide exhibited the lowest binding energies (−6.50 to −6.70 kcal/mol) followed by patchoulol, germacrene D-4-ol, and *epi*-α-muurolol (−6.37 to −6.45 kcal/mol). In addition, nine of the ten sesquiterpenes hydrocarbons bound to the E1-E2-E3 complex, and α-guaiene, α-humulene, and *trans*-β-caryophyllene exhibited the lowest binding energies (−6.32 to −6.38 kcal/mol) followed by α-bulnesene, germacrene D, and δ-cadinene (−6.25 to −6.26 kcal/mol). The sesquiterpenes were accommodated in two consensus sites corresponding to a pocket in the domain II of the E1 protein (six sesquiterpenes) and a pocket in the β-ribbon connector of E2 protein (eleven sesquiterpenes). All EO compounds formed hydrophobic bonds with amino acid residues and five of the top compounds formed hydrogen bonds with amino acids lining the pocket (Figure 3). Table 7 presents the EO compounds with the lowest binding energy with the CHIKV E1-E2-E3 complex.

## 3. Discussion

Cleaning virus-contaminated hands and surfaces is essential for infection control and viral disease prevention [4]. OH-based solutions are utilized as disinfectants to control the transmission of human pathogenic viruses. However, frequent and prolonged use of OH-based disinfectants may be harmful to health and the environment [7,8,9]. EOs in the form of natural products are generally recognized as safe (GRAS) by the FDA (Food and Drug Administration, Silver Spring, MD, USA), and their use is permitted [27]. Many studies have explored using EOs as potential antibacterial and antifungal alternatives to commercial disinfectants [14,28]. In contrast, scientific evidence supporting the potential of EOs as disinfectants against enveloped viruses is very limited. Our study focused on enveloped viruses; studies show that enveloped viruses tend to infect more host species and are more likely to be pandemic than non-enveloped viruses [1,29].

The present study evaluated the virucidal disinfectant activity of solutions containing EO and OH against two surrogate viruses for pathogenic enveloped viruses. The results show that a solution of 12% EO combined with 10% OH reduced up to >4.0 log10 TCID_50_ (100% reduction) of both viruses within 1 min of exposure. In addition, the solutions containing EO without OH also exhibited virucidal action (100% reduction) against both viruses after 10 min (12% EO) and 30 min (3% EO) of exposure. We did not observe a 100% reduction in either virus with the 10% OH solution, but when combined with 12% EO, a strong virucidal activity was observed. It appears that low concentrations of EO and OH are insufficient to inactivate human pathogenic enveloped viruses. Romeo et al. [18] did not observe virucidal activity of a formulation containing 3.3% EO (*Melaleuca alternifolia*) combined with 5.3% ethanol against the coronavirus HCov-OC43 after 30 min of exposure.

To evaluate the virucidal disinfectant activity, we used two enveloped viruses, which differ in the lipid content [30,31] and protein structure [24,26] that comprise the viral envelope. The results indicated that DENV-2 was more sensitive to the action of test solutions than CHIKV. We hypothesized that differences in the viral envelope structure and its hydrophobic/hydrophilic nature might explain the variation in sensitivity. The DENV-2 particle assembles and buds into the endoplasmic reticulum of the infected cells where the envelope is formed. The envelope has 90 head-to-tail dimers of the E protein organized in a herringbone, with the M protein bound at the dimer interface [32]. On the other hand, CHIKV assembles and budding occurs at the cytoplasmic membrane, and the viral envelope comprises the E1 and E2 glycoproteins and a peptide (E3) arranged in trimers to make 80 E1/E2 spikes [27]. A recent study [18] showed differences in the sensitivity of enveloped viruses (human and feline coronaviruses) to treatment with a mixture of tea tree oil and ethanol.

Enveloped viruses enter host cells primarily via endocytosis following attachment to a cellular receptor [2,29,33]. Upon attachment, viruses are engulfed into endosomes where the low pH triggers conformational changes of the envelope proteins to drive fusion of the viral envelope and endosomal membrane. The viral envelope plays an important role in the membrane fusion process [33], and envelope proteins are potential extracellular drug targets with multiple strategies to inhibit entry of the virus into host cells [34]. Studies suggest that EOs could cause the morphological alteration of the viral particle by destroying the viral envelope through interactions between their terpene constituents and viral proteins [11,13]. In silico and in vitro evidence suggests that sesquiterpene hydrocarbons and oxygenated monoterpenes in specific ratios may account for the antiviral action of the EOs [11,12,13]. Recently, we documented a variation in the anti-DENV effect related to variation in oxygenated monoterpene content [25]. We also documented [35] a better in vitro anti-DENV effect of *trans*-β-caryophyllene and geranyl acetate compared to *p*-cymene, limonene, and neral, all of which were identified in the test EO blend.

We performed a primary docking analysis to describe the interactions between the 42 compounds of the EO blend and the envelope proteins of DENV-2 (E) and CHIKV (E1-E2-E3). As in a previous study [25], in the present study, we again found sesquiterpene hydrocarbons and oxygenated monoterpenes showing good binding affinities (−6.7 to −8.6 kcal/mol) with the DENV-2 E protein. These terpenes were accommodated in the βOG pocket and molecules that dock this pocket can block the conformational change of the E protein required for the fusion process [36]. As for CHIKV, seventeen EO compounds docked the E1-E2-E3 glycoprotein complex. Some bound to the E2 protein in a pocket of the β-ribbon connector peptides, which play a role during virus entry, helping to trigger E1 conformational changes during the fusion process [37]. Other EO compounds bound to the E1 protein of CHIKV near the hydrophobic fusion loop, which mediates membrane fusion [37]. According to the docking score values, EO compounds exhibited better binding affinities against DENV-2 than against CHIKV, which could partly explain the differences in sensitivity to the test solutions revealed in the virucidal disinfectant assays.

Little is known about the specific mechanism of action of EOs against enveloped viruses. Mechanisms other than alterations of the envelope protein structure have been proposed [11,12,13]. Being lipophilic, EOs can penetrate the viral envelope and cause membrane disintegration; they can cause viral expansion, which interferes with the attachment process by which viruses gain entry into host cells; moreover, EO components can inhibit host lipid metabolism pathways, which are crucial to ensure the availability of lipids to complete the assembly of new enveloped virions. On the other hand, OH causes protein denaturation and the disruption of the viral envelope [5]. Ethanol (95%, *v*/*v*) has broader and stronger virucidal activity than propanols (75% *v*/*v*); isopropanol, due to its lipophilic nature, interacts favorably with viral envelopes, and glycerol (80% *v*/*v*) and glycerol derivatives have been described as virucidal agents against enveloped viruses [38,39]. We hypothesize that the EO and OH mechanisms mentioned here could be involved in the strong virucidal disinfectant activity of the 12% EO + 10% OH solution.

The results of this study demonstrate that EO alone not only has disinfectant activity, but also shows synergistic activity with OH against two enveloped viruses. This synergistic activity may involve all of the aforementioned mechanisms of action of EOs. Further analysis is needed to investigate the contribution of each EO compound and their additive, synergistic, or antagonistic effects on the disinfectant action of a pure EO.

## 4. Materials and Methods

### 4.1. Plant Material and EO Blend

Pure EOs from seven aromatic plants grown in Colombia were used to prepare an EO blend. Then, a stock solution (6 × 10^6^ µg/mL) of the EO blend was prepared in DMSO and it was used to prepare the test solutions for analyses of the disinfectant activity (Table 1). EOs were distilled from *Cymbopogon martinii* (Roxb.) Will Watson (Poaceae family), *Cymbopogon winterianus* (Java citronella) Jowitt ex Bor (Poaceae family), *Pogostemon cablin* (Blanco) Benth, *Lippia origanoides* Kunth (Verbenaceae family), *Elettaria cardamomum* (L.) Maton (Zingiberaceae family) *Swinglea glutinosa* (Blanco) Merr (Rutaceae family), and *Citrus* × *aurantium* L. (Rutaceae family). The plants were grown in the experimental plots at the Agroindustrial Pilot Complex of the National Center for Agroindustrialization of Aromatic and Medicinal Tropical Vegetal Species (CENIVAM) in the Industrial University of Santander (Bucaramanga, Colombia). The taxonomic identification of these plants was performed at the Colombian National Herbarium (Bogotá, Colombia), where their vouchers were placed. EOs were obtained through the hydrodistillation (2 h) of plant leaves and stems on a Clevenger apparatus as described elsewhere [40,41].

### 4.2. Chemical Composition of the EO Blend

The analysis of the EO blend was performed by gas chromatography using mass spectrometric (GC/MS) and flame ionization detection (GC/FID) systems. Previous studies described the conditions of the process and data analysis [25,42,43]. Before the analysis, the EO blend was dissolved in dichloromethane (1 mL). *n*-Tetradecane (0.5 µL) was added as an internal standard. The injection volume was 2 µL in split mode (30:1). A 6890 Plus Gas Chromatograph (Agilent Technologies, AT, Palo Alto, CA, USA) equipped with a mass selective detector MSD 5975 (Electron ionization, EI, 70 eV), (AT, Palo Alto, CA, USA), a 7863 automatic injector, and an MSChemStation G1701DA data system (AT, Palo Alto, CA, USA) were used. The identification of EO compounds was accomplished by the comparison of their linear retention indices (LRIs) with those of standard compounds, and by the comparison of their mass spectral fragmentation patterns with those described in the literature and databases [20,21,22,23,24].

### 4.3. Preparation of the Test Solutions

Pure EOs from seven aromatic plant species were mixed in various proportions to obtain an EO blend using dimethyl sulfoxide (DMSO) as the solvent. The EO blend was mixed with the desired amount of an OH mixture (ethanol ca. 70%; isopropanol + glycerol ca. 2.5%) in a glass vial to give five different percentage ratios of EO/OH. Each solution was stirred using a vortex mixer until complete mixing took place. In addition, the EO blend was diluted to give solutions of 12% and 3%, and the OH mixture was diluted in water to prepare a 10% solution.

### 4.4. Cells and Viruses

Vero cells (African green monkey kidney cells; CCL-81™. ATCC, Manassas, VA, USA) were cultured in minimum essential medium (MEM) supplemented with 10% fetal bovine serum (FBS) and 1% antibiotic at 37 °C in a humidified atmosphere of 5% CO_2_. DENV-2 NGC (CDC, San Juan, Puerto Rico) was propagated in C6/36 *Aedes albopictus* cells (Pedro Kourí Institute for Tropical Medicine, La Habana, Cuba). CHIKV, a local strain isolated from a patient in our laboratory [44], was propagated in Vero cells. Both viruses were titrated using a protocol of the median tissue culture infectious dose (TCID_50_)—Spearman Karber assay [45].

### 4.5. Cytotoxicity Controls

As EO and OH can cause cytotoxic effects, the test solutions were first evaluated in Vero cells without the addition of virus. Briefly, the test solution was serially diluted, and an aliquot was added to cells seeded in 96-well plates. Following 1 h of incubation at 37 °C, the solution was discarded by washing and the cells were overlaid with fresh culture medium and incubated for 72 h at 37 °C and 5% CO_2_. Next, the cell viability was determined by staining with crystal violet, as in a previous study [25]. Briefly, 100 µL of 0.05% crystal violet solution was added to cells for 20 min at room temperature. After washing, the plates were aspirated and allowed to air dry at room temperature, and 200 µL of methanol was added to each well for 20 min. The optical density at 570 nm in each well was measured on a microplate reader (570 nm) to quantify crystal violet staining.

### 4.6. Evaluation of Virucidal Disinfectant Activity

The quantitative suspension test was used following the German *DVV*/RKI *guideline* [13,18]. The test was performed in five intervals (1, 5, 10, 20, and 30 min) of exposure of the virus with the test solution with fixed amounts of DENV-2 (8.4 log10 TCID_50_/mL) and CHIKV (7.8 log10 TCID_50_/mL). Briefly, 10 µL of a virus preparation was mixed with 80 µL of solution and 10 µL of water, and a virus control with 90 µL of water without test solution was included. At the end of the exposure times, 900 µL of ice-cold culture medium was added to each mixture and immediately diluted 10-fold to determine viral infectivity using end-point dilution titration. Vero cells were seeded in 96-well plates for 24 h at 37 °C under 5% CO_2_ and infected with serial dilutions of treated DENV-2 and treated CHIKV in triplicate on a logarithmic scale at base 10. Noninfected cells were included as controls. The plates were incubated at 37 °C and 5% CO_2_ for five days. After washing, the plates were aspirated and allowed to air dry at room temperature, and the crystal violet dye uptake was determined as described above. The quantity of virus was calculated as TCID_50_ (log10) per milliliter by the Spearman–Karber method [45].

### 4.7. Docking Analysis

Three-dimensional structures of DENV-2 E protein (PDB ID: 10AN) and the CHIKV E1-E2-E3 complex (PDB ID: 3N42) were downloaded from the Protein Data Bank. Structures of chemical constituents of the EO blend were retrieved from the PubChem (https://pubchem.ncbi.nlm.nih.gov/ (accessed on 13 April 2023)) database. The preparation of the target and ligands and molecular docking analyses were carried out using AutoDock Vina (Version 1.5.6, La Jolla, CA, USA), as described in a previous study [25]. The optimized protein structure was saved in the PDBQT file format for docking analysis. Default parameters were used, and the search exhaustiveness parameter was set to 100. For each ligand, 27 docked conformations were generated using global docking simulations. Three simulations were performed for each ligand–protein pair using seeds 6, 12, and 18. The average docking scores for each protein approximated the binding free energy. Discovery Studio Visualizer v21.1.0.20298 was used to view the ligand–protein interactions.

### 4.8. Statistical Analyses

A one-way ANOVA and a Tukey–Kramer post hoc test of viral titer values were used to compare the virucidal effect of each test solution, adopting a significance level of 0.05. The data were analyzed using GraphPad Prism software (version 8.0, San Diego, CA, USA).

## 5. Conclusions

The inadequate and inappropriate use of OH-based and other disinfectants has been associated with harmful effects on humans and the environment. There is an urgent need to develop safe and environmentally friendly disinfectants to minimize adverse effects. The data from this study provide a first step in defining the potential utility of EOs as disinfectants to control the transmission of human pathogenic enveloped viruses. We conclude that a solution containing 12% of a mixture of seven EOs and 10% of a mixture of OHs (ethanol, isopropanol, and glycerol) is highly effective for the inactivation of DENV-2 and CHIKV, which could be extended to enveloped viruses of similar structure that are transmitted person-to-person. In addition to reducing virus titers to 100%, the solution acts within one minute, making it practical for use in environments where rapid disinfection is needed. The hydrocarbon sesquiterpenes, oxygenated sesquiterpenes, and oxygenated monoterpenes present in the EO blend showed binding affinities for DENV-2 and CHIKV envelope proteins, suggesting that these types of terpenes could act as inhibitors of virus adsorption and entry into host cells. Further analysis is needed to better define the potential of EOs as virucidal disinfectant alternatives to commercial OH-based disinfectants.

## Figures and Tables

**Figure 1 molecules-28-04156-f001:**
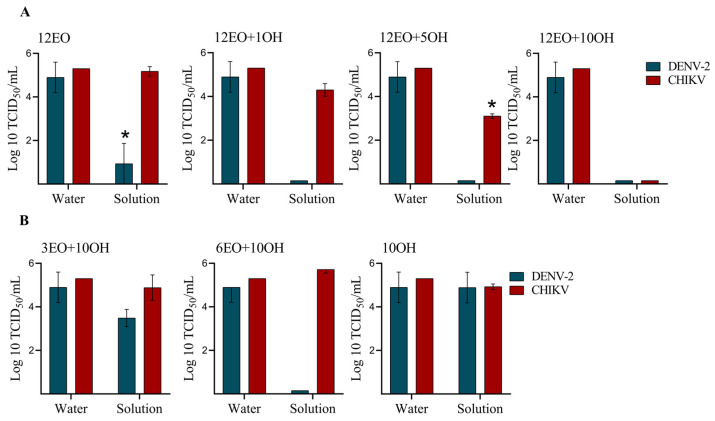
Comparison of virucidal disinfectant activities of the test solutions against DENV-2 and CHIKV within 1 min of exposure. Table 1 presents the content of essential oil (EO) and alcohol (OH) in each solution. (**A**) Solutions based on 12% EO with increasing alcohol concentration (1%, 5% and 10%). (**B**) Solutions based on 10% alcohol with increasing EO concentration (3% and 6%). The residual infectivity was determined by virus titer using the TCID_50_ assay. Data are averages ± SDs from three independent assays in triplicate. * One-way ANOVA: DENV-2: F_9,17_ = 21.64; and CHIKV: F_7,31_ = 14.57, *p* < 0.001; Tukey’s post hoc test, *p* < 0.001.

**Figure 2 molecules-28-04156-f002:**
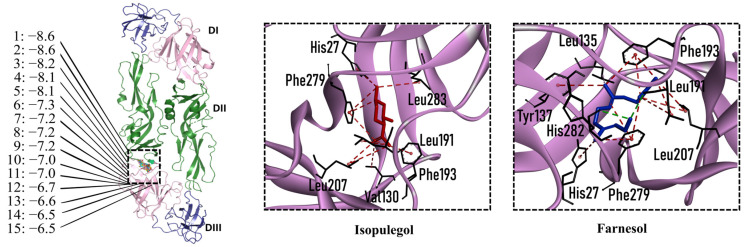
Docked poses of representative EO compounds in complex with DENV-2 E protein into the βOG pocket. Interacting amino acids are shown as black sticks; hydrogen bonding interactions are depicted as green dotted lines and hydrophobic interactions are depicted as red dotted lines. Compounds: 1, δ-cadinene; 2, α-guaiene; 3, γ-cadinene; 4, α-bulnesene; 5, carvacrol; 6, *epi*-α-muurolol; 7, γ-terpinene; 8, limonene; 9, *p*-cymene; 10, germacrene D; 11, thymol; 12, farnesol; 13, isopulegol; 14, α-cadinol; 15, α-eudesmol.

**Figure 3 molecules-28-04156-f003:**
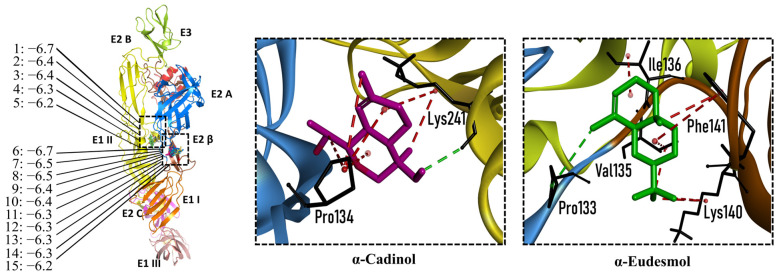
Docked poses of representative EO compounds in complex with the E3-E2-E1 protein complex of CHIKV. Interacting amino acids are shown as black sticks; hydrogen bonding interactions are depicted as green dotted lines and hydrophobic interactions are depicted as red dotted lines. Compounds into a pocket of E1 domain II: 1, α-cadinol; 2, α-guaiene, 3: *epi*-α-muurolol; 4: *epi*-α-cadinol; 5, γ-cadinene. Compounds into a pocket of the β-ribbon connector of E2: 6, α-eudesmol; 7, caryophyllene oxide; 8, patchoulol; 9, germacrene D-4-ol; 10, α-humulene; 11, *trans*-β-caryophyllene; 12, α-bulnesene; 13, germacrene D; 14, δ-cadinene; 15, α-muurolene.

**Table 1 molecules-28-04156-t001:** Solutions tested for virucidal disinfectant activity.

No.	Content	DMSO, %	Identifier
1	12% EO + 1% OH	7.9	12EO + 1OH
2	12% EO + 5% OH	7.6	12EO + 5OH
3	12% EO + 10% OH	7.2	12EO + 10OH
4	6% EO + 10% OH	7.2	6EO + 10OH
5	3% EO + 10% OH	1.8	3EO + 10OH
6	12% EO	8.0	12EO
7	3% EO	2.0	3EO
8	10% OH	-	10OH

EO is a blend of pure EOs from seven aromatic plants. OH is a mixture of ethanol (ca. 70%) and other OHs (ca. 2.5%: isopropanol and glycerol). Dimethyl sulfoxide (DMSO) was used as an EO solvent and the final concentration in the solution is presented.

**Table 2 molecules-28-04156-t002:** Viability of Vero cells exposed to the test solutions for 72 h.

Solution	Dilution/Percentage of Viability		
	1:10	1:100	1:1000
12EO	80 ± 26	90 ± 13	100 ± 10
12EO + 1OH	80 ± 11	92 ± 8.4	96 ± 6.0
12EO + 5OH	80 ± 16	93 ± 7.3	97 ± 2.7
12EO + 10OH	80 ± 25	90 ± 10	100 ± 10
6EO + 10OH	92 ± 9.2	96 ± 7.1	98 ± 3.3
3EO + 10OH	92 ± 8.9	95 ± 9.1	95 ± 7.3
3EO	100 ± 0.0	94 ± 8.9	90 ± 33
10OH	94 ± 5.4	90 ± 8.8	90 ± 10
Acetic acid	0.0 ± 3.2	80 ± 32	93 ± 4.9

Acetic acid (5%) was used as a virucidal agent. Data are averages ± SDs from three independent assays in triplicate.

**Table 3 molecules-28-04156-t003:** Disinfectant activity of the test solutions after one minute of virus exposure.

Solution	DENV-2: TCID_50_/mL			CHIKV: TCID_50_/mL		
	Log10	RF	% R	Log10	RF	% R
Water	3.8 ± 0.4 ^†^ and 5 ± 1.2			5.3 ± 0.5	-	-
Acetic acid	0.0	4.9 ± 0.0	100	0.0	5.3 ± 0.5	100
12EO	1 ± 1.6	3.9 ± 0.4	81.6	5.3 ± 0.5	0.0	0.0
12EO + 10OH	0.0	4.9 ± 0.0	100	4.3 ± 0.7	0.9 ± 0.7	18.8
12EO + 5OH	0.0	4.9 ± 0.0	100	3.1 ± 0.2 *	2.2 ± 0.2	41.5
12EO + 10OH	0.0	4.9 ± 0.0	100	0.0	5.3 ± 0.0	100
6EO + 10OH	0.0	4.9 ± 0.0	100	5.7 ± 0.2	0.0	0.0
3EO + 10OH	3.5 ± 0.7	0.5 ± 0.7	28.5	5 ± 1.0	0.4 ± 1.0	9.5
3EO	3.1 ± 0.3	0.7 ± 0.6	18.4	ND	ND	-
10OH	4.2 ± 0.0	0.7 ± 0.0	0.0	5.3 ± 0.5	0.0	0.0

^†^ Virus control for the 3EO solution. RF: reduction factor, Log10 TCID_50_ virus control—Log 10 TCID_50_ treated virus; % R, percentage reduction in virus titer relative to virus control; 0.0, the virus was not detected. ND, not determined. Data are averages ± SDs from three independent assays in triplicate. * One-way ANOVA: DENV-2: F_9,17_ = 21.64; and CHIKV: F_7,31_ = 14.57, *p* < 0.001; Tukey’s post hoc test, *p* < 0.001.

**Table 4 molecules-28-04156-t004:** Virucidal disinfectant activity of essential oil solutions without alcohol according to time of exposure.

Solution	DENV-2: Log10 TCID_50_/mL				CHIKV: Log10 TCID_50_/mL			
	Control	Treated	RF	R, %	Control	Treated	RF	R, %
12EO								
5 min	5.4 ± 0.2	0.0	5.4 ± 0.2	100	5.7 ± 0.7	3.0 ± 0.7 *	2.7 ± 0.1	47.3
10 min	5.0 ± 1.0	0.0	5.0 ± 1.5	100	4.8 ± 0.4	0.0	4.8 ± 0.3	100
20 min	5.4 ± 0.5	0.0	5.4 ± 0.5	100	4.9 ± 0.1	0.0	4.0 ± 0.1	100
30 min	4.5 ± 0.9	0.0	4.5 ± 0.9	100	4.6 ± 0.5	0.0	4.6 ± 0.5	100
3EO								
5 min	4.0 ± 0.1	3.0 ± 0.6	0.9 ± 0.5	25	6.1 ± 0.6	5.4 ± 0.5	0.6 ± 0.6	11.4
10 min	5.0 ± 1.5	3.3 ± 0.6	1.0 ± 0.7	26.6	5.1 ± 0.2	5.1 ± 0.4	0.0	0.0
20 min	3.7 ± 0.1	2.1 ± 0.8	0.4 ± 1.0	43.2	6.1 ± 0.3	3.3 ± 0.7 *	2.8 ± 0.8	45.9
30 min	3.7 ± 0.4	0.0 ± 1.4	3.7 ± 0.1	100	5.0 ± 1.5	0.0 ± 0.0	5.0 ± 1.5	100

RF: reduction factor; R, %: percentage reduction in virus titer; 0.0, the virus was not detected. Data are averages ± SDs from three independent assays in triplicate. * One-way ANOVA: DENV-2: F_7,16_ = 20.27; and CHIKV: F_7,14_ = 46, *p* < 0.001; Tukey’s post hoc test, *p* < 0.001.

**Table 5 molecules-28-04156-t005:** Chemical composition of the EO blend.

No.	Compound	Type	Linear Retention Indices		GC/FID Relative Peak Area, %
			Exp.	Lit.	
1	α-Pinene *	MH	934	932 ^a^	0.1
2	6-Methyl-hept-5-en-2-one	OC	984	985 ^a^	0.1
3	β-Myrcene *	MH	989	990 ^a^	0.2
4	*p*-Cymene *	MH	1026	1027 ^a^	0.8
5	Limonene *	MH	1031	1029 ^a^	2.0
6	1,8-Cineole *	OM	1036	1034 ^a^	0.3
7	*trans*-β-Ocimene	MH	1047	1050 ^a^	0.9
8	γ-Terpinene *	MH	1060	1059 ^a^	0.2
9	Linalool *	OM	1100	1096 ^a^	2.2
10	Citronellal *	OM	1157	1153 ^a^	22.6
11	Isopulegol	OM	1165	1155 ^a^	0.3
12	*n*-Decanal	OC	1207	1201 ^a^	0.1
13	Citronellol	OM	1220	1233 ^a^	14.1
14	Neral	OM	1241	1242 ^b^	0.4
15	Geraniol *	OM	1258	1255 ^b^	35.4
16	Geranial *	OM	1271	1270 ^b^	0.8
17	Thymol *	PhC	1292	1290 ^a^	1.1
18	Carvacrol *	PhC	1301	1298 ^a^	2.1
19	Citronellyl acetate	OM	1346	1350 ^a^	2.5
20	Eugenol	PhC	1354	1356 ^a^	0.4
21	Geranyl acetate	OM	1377	1379 ^a^	4.7
22	β-Elemene	SH	1396	1389 ^a^	1.0
23	*trans*-β-Caryophyllene *	SH	1432	1428 ^d^	0.9
24	α-Guaieno	SH	1444	1440 ^b^	0.1
25	α-Humulene *	SH	1468	1465 ^d^	0.2
26	γ-Muurolene	SH	1484	1478 ^a^	0.1
27	Germacrene D *	SH	1492	1481 ^a^	1.5
28	α-Muurolene	SH	1506	1500 ^a^	0.4
29	α-Bulnesene	SH	1511	1509 ^a^	0.1
30	γ-Cadinene	SH	1523	1513 ^a^	0.3
31	δ-Cadinene	SH	1526	1522 ^a^	1.1
32	Elemol	OS	1557	1548 ^a^	0.9
33	*trans*-Nerolidol *	OS	1565	1561 ^a^	0.2
34	Germacrene D-4-ol	OS	1578	1574 ^a^	0.7
35	Caryophyllene oxide *	OS	1586	1582 ^a^	0.1
36	epi-α-Cadinol	OS	1653	1650 ^c^	0.1
37	epi-α-Muurolol	OS	1655	1642 ^c^	0.2
38	α-Cadinol	OS	1667	1653 ^c^	0.2
39	α-Eudesmol	OS	1669	1659 ^c^	0.1
40	Patchoulol	OS	1691	1660 ^b^	0.1
41	Farnesol	OS	1719	1723 ^b^	0.1
42	Neryl hexanoate	OM	1750	1732 ^c^	0.2
1. Monoterpenoids					87.7
1.1 Monoterpene hydrocarbons (MH)					4.2
1.2 Oxygenated monoterpenes (OM)					83.5
Alcohols					52
Acetates					7.2
Aldehydes					23.8
Others (ethers, esters, epoxides)					0.5
2. Sesquiterpenoids					8.4
2.1 Sesquiterpene hydrocarbons (SH)					5.7
2.2 Oxygenated sesquiterpenes (OS)					2.7
Alcohols					2.6
Others (Oxides)					0.1
3. Phenolic compounds (PhC) (Thymol, carvacrol, eugenol)					3.6
4. Other oxygenated compounds (OC) (*n*-Decanal, 6-methyl-5-hepten-2-one)					0.2

LRI, linear retention indices calculated using *n*-alkanes C_8_–C_25_ mixture on the DB-5MS (non-polar) column. Exp., experimental. Lit., literature: ^a^ [20]; ^b^ [21]; ^c^ [22]; ^d^ [23]. * Use of standard compounds.

**Table 6 molecules-28-04156-t006:** EO compounds with binding affinity to the E protein of DENV-2.

Compound	Structural Formula	Amino Acid Residues. H-Bond in Bold Font	Kcal/mol
Isopulegol	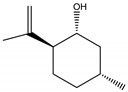	Thr189, Leu191, Phe193, Leu207, Phe279	−6.70 ± 0.6
Farnesol	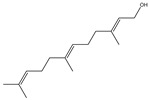	Thr48, Leu135, Tyr137, **Gly190**, **Leu191**, Phe193, Leu207, Phe279, **His282**, Leu283	−6.57 ± 0.6
α-Cadinol	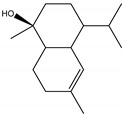	**Thr48, Tyr137**, Thr189, Leu191, Phe193, Leu207, Phe279	−6.54 ± 0.5
Eugenol	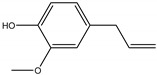	Thr48, Leu135, Thr189, Leu191, Phe193, Phe279, **Gly281, His282**	−6.54 ± 0.5
α-Eudesmol	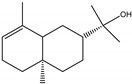	Thr48, Val130, Phe193, Leu207, Phe279, Leu283	−6.50 ± 0.6

The twelve compounds with a strong binding energy (−7.03 to −8.61 kcal/mol) and another eight compounds with a weak binding energy (−6.0 to −6.7 kcal/mol) were reported in a previous study [25].

**Table 7 molecules-28-04156-t007:** EO compounds with the lowest binding energy with the E1-E2-E3 glycoprotein complex of CHIKV.

Compound	Structural Formula	Protein	Amino Acid Residues. H-Bond in Bold Font	Kcal/mol
α-Cadinol	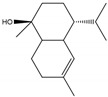	E1, domain II	Asn39, Thr42, Pro133, Pro134, **Lys241**, Leu244	−6.70 ± 0.2
α-Eudesmol	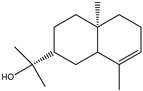	E2, β-ribbon connector	**Pro133**, Val135, Ile136, Lys140, Phe141	−6.70 ± 0.3
Caryophyllene oxide	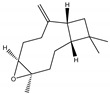	E2, β-ribbon connector	**Arg104**, Pro133, Val135, Ile136, Lys40, Phe141	−6.50 ± 0.31
Patchoulol	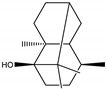	E2, β-ribbon connector	Pro133, Ile136, Lys140, Phe141, **Asp43**	−6.45 ± 0.2
α-Guaiene	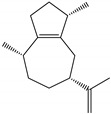	E1, domain II	Pro134, Lys241, Tyr242, Lys245	−6.38 ± 0.2
Germacrene D-4-ol	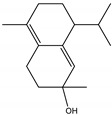	E2, β-ribbon connector	Thr42, Pro134, Lys241, Leu244, Lys245, **Asn39**	−6.38 ± 0.2
*epi*-α-Muurolol	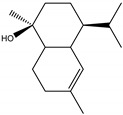	E1, domain II	Pro134, Lys241, Leu244, Lys245	−6.37 ± 0.2
α-Humulene	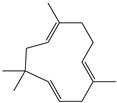	E2, β-ribbon connector	Ile136, Phe141	−6.37 ± 0.3
*trans*-β-Caryophyllene	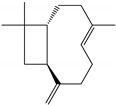	E2, β-ribbon connector	Ile136, Phe141, Arg144	−6.32 ± 0.1

## Data Availability

All data, tables, and figures are originals.

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
