# Peer review of "Comparative Virucidal Activities of Essential Oils and Alcohol-Based Solutions against Enveloped Virus Surrogates: In Vitro and In Silico Analyses"

_molecules, 2023, doi:10.3390/molecules28104156_

Round 1

Reviewer 1 Report

The manuscript entitled '' Comparative virucidal activities of essential oils and alcohol-based solutions against enveloped virus surrogates: in vitro and in silico analyses'' is an interesting and important study. However, the authors need to make some changes.

·       In the introduction, the authors should cite more recent and relevant references to cover the relevant literature and support their study.

·       Please check the abbreviations throughout the manuscript. It would be best to introduce the acronym when the whole word appears the first time in the text and then use only the abbreviations.

·       The resolution of some Figures is unsatisfactory. Could you please increase the resolution and use a larger font size, not a large figure size only?

·       Please provide a brief illustration of Solutions, groups, and EO at the beginning of the results as the material and methods section at the end of the manuscript.

·       The structural formula of the compounds was drawn by authors or from websites.

·       The discussion needs some improvement. Please, clarify the advantage of this study over others. Please highlight and emphasize the novelty of this study. 

·       In the material and methods section, cite a relevant reference in Virucidal assays.

-        Authors need to correct some grammatical mistakes.

Author Response

Responses to Reviewer 1
We thank the reviewer for his/her valuable suggestions and for recognizing the value of the
presented data. We incorporated all the suggestions and we are confident of having improved
the quality of our manuscript. We present here our replay to each of comment separately.
Comment 1:
“In the introduction, the authors should cite more recent and relevant references to
cover the relevant literature and support their study”.
We agree with the reviewer. We have included more recent and relevant references, which
cover the relevant literature and support our study:
Reference 1 (line 33): Bhadoria, P.; Gupta, G.; Agarwal, A. Viral pandemics in the past two
decades: an overview.
J Family Med Prim Care. 2021, 10, 2745-2750.
Reference 4 (line 39): Suchomel, M.; Eggers, M.; Maier, S.; Kramer, A.; Dancer, S.J.; Pittet,
D. Evaluation of world health organization-recommended hand hygiene formulations.
Emerg Infect Dis, 2020, 26, 2064-2068.
Reference 9 (line 44): Jing, J.L.; Pei, Yi.; T, Bose, R.J.C.; McCarthy, J.R.; Tharmalingam,
N.; Madheswaran, T. Hand Sanitizers: A review on formulation aspects, adverse effects,
and regulations.
Int J Environ Res Public Health. 2020, 17, 1-17.
Reference 14 (line 56): Bolouri, P.; Salami, R.; Kouhi, S.; Kordi, M.; Asgari-Lajayer, B.;
Hadian, J.; Astatkie, T. Applications of essential oils and plant extracts in different
industries.
Molecules. 2022, 27, 1-17.
Reference 15 (line 56): Pizzo, J.S.; Visentainer, J.V.; da Silva, ALBR.; Rodrigues, C.
Application of essential oils as sanitizer alternatives on the postharvest washing of fresh
produce
. Food Chem. 2023, 407, 1-17.
Reference 16 (line 56): Maurya, A.; Prasad, J.; Das and, S; Dwivedy, A.K. Essential oils and
their application in food safety.
Front. Sustain. Food Syst. 2021, 5, 1-25.
Comment 2:
“Please check the abbreviations throughout the manuscript. It would be best to introduce
the acronym when appears the first time in the text and then use only the abbreviations”.
Thank for the comment. We have checked that the acronyms and abbreviations appear after
the whole definition or word throughout the manuscript.
Comment 3:
The resolution of some Figures is unsatisfactory. Could you please increase the
resolution and use a larger font size, not a large figure size only
?”
Thank for the comment. We have improved the resolution, used larger font size and
improved the color of figures 1, 2 and 3.

Comment 4:
Please provide a brief illustration of Solutions, groups, and EO at the beginning of the
results as the material and methods section at the end of the manuscript”.
We have provided a brief illustration of solutions, groups, and EO at the beginning of the
results section (lines 70 - 74) …” ….. A single EO blend was used, which contained pure
EOs from seven Colombian aromatic plants. In addition, an OH preparation was used, which
contained ethanol (70%) and a mixture (25%) of isopropanol and glycerol. Eight solutions
containing EO (3, 6 and 12%) combined or not with the OH preparation (1, 5 and 10%) were
analysed….”
Comment 5:
“The structural formula of the compounds was drawn by authors or from websites”
We downloaded the structural formula of the compounds from NIST Chemistry WebBook.
In the new version of the manuscript, we have drawn the formulas using the software
Chemdraw, 9.0.
Comment 6:
The discussion needs some improvement. Please, clarify the advantage of this study over
others. Please highlight and emphasize the novelty of this study

We agree with the reviewer. We have improved the discussion section clarifying and
emphasizing the advantages and novelty of our study.
3. Discussion
Cleaning virus-contaminated hands and surfaces is essential to
infection control and viral disease prevention [4]. OH
-based solutions
are utilized as disinfectants to control the transmission of human
pathogenic viruses. However, frequent and prolonged use of OHbased disinfectants may be harmful to health and the environment [7-
9]. EOs in the form of natural products are generally recognized as safe
(GRAS) by the FDA (Food and Drug Administration, U.S.A.), and their
use is permitted [23]. Many studies have explored using EOs as
potential antibacterial and antifungal alternatives to commercial
disinfectants [14,24]. In contrast, scientific evidence supporting the
potential of EOs as disinfectants against enveloped viruses is very
limited. Our study focused on enveloped viruses, studies show
that enveloped viruses tend to infect more host species and are more
likely to be pandemic than non-enveloped viruses [1, 25].
The present study evaluated the virucidal disinfectant activity of
solutions containing EO and OH against two surrogate viruses for
pathogenic enveloped viruses. The results show that a solution of 12%
EO combined with 10% OH reduced up to > 4.0 log10 TCID
50 (100%
reduction) of both viruses within 1 min of exposure. In addition,
solutions containing EO without OH also exhibited virucidal action
(100% reduction) against both viruses after 10 min (12% EO) and 30 min

(3% EO) of exposure. We did not observe a 100% reduction in either
virus with the 10% OH solution, but when combined with 12% EO, a
strong virucidal activity was observed. It appears that low
concentrations of EO and OH are insufficient to inactivate human
pathogenic enveloped viruses. Romeo et al. [18] did not observe
virucidal activity of a formulation containing 3.3% EO (
Melaleuca
alternifolia
) combined with 5.3% ethanol against the coronavirus HCovOC43 after 30 min of exposure.
To evaluate the virucidal disinfectant activity, we have used two
enveloped viruses, which differ in the lipid content [26,27] and protein
structure [20,22] that comprise the viral envelope. The results indicated
that DENV-2 was more sensitive to the action of test solutions than
CHIKV. We hypothesized that differences in the viral envelope
structure and its hydrophobic/hydrophilic nature might explain the
variation in sensitivity. The DENV-2 particle assembles and buds into
the endoplasmic reticulum of the infected cells where the envelope is
formed. The envelope has 90 head-to-tail dimers of the E protein
organized in a herringbone, with the M protein bound at the dimer
interface [28]. On the other hand, CHIKV assembles and budding
occurs at the cytoplasmic membrane, and the viral envelope comprises
the E1 and E2 glycoproteins and a peptide (E3) arranged in trimers to
make 80 E1/E2 spikes [23]. A recent study [18], showed differences in
the sensitivity of enveloped viruses (human and feline coronaviruses)
to treatment with a mixture of tea tree oil and ethanol.
Enveloped viruses enter host cells primarily via endocytosis
following attachment to a cellular receptor [2,25,29]. Upon attachment,
viruses are engulfed into endosomes where the low pH triggers
conformational changes of the envelope proteins to drive fusion of the
viral envelope and endosomal membrane. The viral envelope plays an
important role in the membrane fusion process [29], envelope proteins
are potential extracellular drug targets with multiple strategies to
inhibit entry of the virus into host cells [30]. Studies suggest that EOs
could cause morphological alteration of the viral particle by destroying
the viral envelope through interactions between their terpene
constituents and viral proteins [11,13].
In silico and in vitro evidence
suggest that sesquiterpene hydrocarbons and oxygenated
monoterpenes in specific ratios may account for the antiviral action of
the EOs [11-13]. Recently, we have documented a variation in the antiDENV effect related to variation in oxygenated monoterpene content
[21]. We have also documented [31] a better
in vitro anti-DENV effect
of
trans-β-caryophyllene and geranyl acetate compared to p-cymene,
limonene and neral, all of which were identified in the test EO blend.
We performed a primary docking analysis to describe the
interactions between the 42 compounds of the EO blend and the
envelope proteins of DENV-2 (E) and CHIKV (E1-E2-E3). As in a
previous study [21], in the present study, we again found sesquiterpene
hydrocarbons and oxygenated monoterpenes showing good binding
affinities (-6.7 to -8.6 kcal/mol) with DENV-2 E protein. These terpenes
were accommodated in the βOG pocket and molecules that dock this
pocket can block the conformational change of the E protein required

for the fusion process [32]. As for CHIKV, seventeen EO compounds
docked the E1-E2-E3 glycoprotein complex, some bound to the E2
protein in a pocket of the β-ribbon connector peptides, which play a
role during virus entry helping to trigger E1 conformational changes
during the fusion process [33]. Other EO compounds bound to the E1
protein of CHIKV near the hydrophobic fusion loop, which mediates
membrane fusion [33]. According to the docking score values, EO
compounds exhibited better binding affinities against DENV-2 than
against CHIKV, which could partly explain the differences in
sensitivity to the test solutions revealed in the virucidal disinfectant
assays.
Little is known about the specific mechanism of action of EOs
against enveloped viruses. Mechanisms other than alterations of the
envelope protein structure have been proposed [11-13]. Being
lipophilic, EOs can penetrate the viral envelope and cause membrane
disintegration; they can cause viral expansion, which interferes with
the attachment process by which viruses gain entry into host cells; and
EO components can inhibit host lipid metabolism pathways, which are
crucial to ensure the availability of lipids to complete the assembly of
new enveloped virions. On the other hand, OH cause protein
denaturation and disruption of the viral envelope [5]. Ethanol (95%,
v/v) has broader and stronger virucidal activity than propanols (75%
v/v); isoproponol, due to its lipophilic nature, interacts favorably with
viral envelopes; and glycerol (80% v/v) and glycerol derivatives have
been described as virucidal agents against enveloped viruses [34,35].
We hypothesize that the EO and OH mechanisms mentioned here
could be involved in the strong virucidal disinfectant activity of the
12%EO +10%OH solution.
Results of this study demonstrate that EO alone not only has
disinfectant activity, but also shows synergistic activity with OH
against two enveloped viruses. This synergistic activity may involve all
of the aforementioned mechanisms of action of EOs. Further analysis is
needed to investigate the contribution of each EO compound and their
additive, synergistic or antagonistic effects on the disinfectant action of
a pure EO.
Comment 7:
“In the material and methods section, cite a relevant reference in Virucidal assays”.
We agree with the reviewer. We have included two references:
Reference 18 (line 350): Romeo, A.; Iacovelli, F.; Scagnolari, C.; Scordio, M.; Frasca, F.;
Condò, R.; Ammendola, S.; Gaziano, R.; Anselmi, M.; Divizia, M.; Falconi, M. Potential use
of tea tree oil as a disinfectant agent against coronaviruses: a combined experimental and
simulation study.
Molecules. 2022, 27, 1-19.
Reference 44 (Line 362): Ramakrishnan, M.A. Determination of 50% endpoint titer using a
simple formula.
WJV, World J Virol. 2016, 5, 85-6.
Comment 9:
“Authors need to correct some grammatical mistakes.”
Thanks for the comment. We have corrected grammatical mistakes and have asked an
English speaker to edit the manuscrip

Reviewer 2 Report

This manuscript describes virucidal activity of an essential oil (EO) blend with or without alcohol by using two enveloped RNA viruses, dengue and chikungunya viruses (DENV-2 and CHIKV, respectively).  The authors also investigated the binding affinities of EO components to the envelope proteins of these viruses in silico and speculated the possible mechanism of virucidal effect of EO.

1. There are previous publications that demonstrated virucidal or antiviral activities of EOs from various origin as reviewed in the reference 9.  Please clearly describe what is new in this manuscript.  Is the essential oil blend unique or the virus the authors used for the virucidal activity test unprecedented?

2. The authors experimentally demonstrated the virucidal activity of an EO blend with DENV-2 and CHIKV.  However, the contribution of each EO component to the virucidal activities by binding to viral envelope protein is only a speculation based on in silico modeling.  These EO components are commercially available.  Including the experimental results that support their hypothesis by using any of these EO components would significantly increase the value of the findings of this manuscript.

3. If the virucidal effect of EO is based on the specific interactions between the EO components and viral envelope proteins I have following questions:

1)    Is the virucidal activity of EO universal to many envelope viruses like alcohol, considering the variety of envelope protein(s) that is unique to each virus?

2)    If viral proteins, not the lipid envelope, are the target molecules of EO components for their virucidal activity why the authors focused on only envelope viruses?

I sense some confusion in the arguments of the authors.  If the authors believe the interactions between the EO components and specific viral proteins, not the envelopes, are the mechanism behind the virucidal activity of EO why the authors can argue EOs can be non-specific disinfectant such as alcohol? 

Minor points:

·      Line 30: “influenza virus H1N1” should be “influenza A virus.”

·      Table 2: Please clearly indicate the unit of numbers in the table.  Are they survival rate in percent?

·      Table 6 and 7: They did not appear in the text.  They must be removed if not needed.

·      Line 211: Virus envelope does not “mediate virus binding to the cell surface …”

·      Line 213: “…differ in the lipid content and…”: This statement requires reference.

·      Line 296: Did the authors mix several EOs in various proportions for the experiments?  Does this mean the authors used various EO blends, not a single mixture, during the study?  The authors need clearer definition of the EO blend (or blends) for the sake of reproducibility.

·      Line 330: Did the authors use only water in the control while the EO blend was prepared in DMSO?

·      Line 334: What does it mean “… with threefold serial dilutions of…on a logarithmic scale at base 10”?  Did the authors make 3-fold dilutions or 10-fold dilutions?

Author Response

 Responses to Reviewer 2
We thank the reviewer for his/her valuable suggestions and for recognizing the value of the
presented data. We incorporated all the suggestions and we are confident of having improved
the quality of our manuscript. We present here our replay to each of comment separately.
Comment 1:
There are previous publications that demonstrated virucidal or antiviral activities of EOs
from various origin as reviewed in the reference 9. Please clearly describe what is new in
this manuscript. Is the essential oil blend unique or the virus the authors used for the
virucidal activity test unprecedented?”
We agree with the reviewer that both virucidal and antiviral activity of EOs is well
documented. However, those studies support the concept that EOs can be used as starting
points for drug discovery to prevent and treat viral infections (references 11-13). EOs have
applications in industries other than pharmaceuticals, including the cleaning-products
industry. Cleaning virus-contaminated surfaces is a crucial part of infection control and viral
disease prevention. Several studies have explored the use of EOs as potential antibacterial
and antifungal alternatives to commercial disinfectants. In contrast, the scientific evidence
supporting EOs as disinfectants against enveloped viruses is very limited. To clean viruscontaminated surfaces, alcohol-based disinfectants are commonly used, and there is
sufficient evidence of their harmful effects on humans, animals and the environment
(references 7-9). Mitigation strategies are required to reduce these effects. Our study presents
evidence supporting the potential use of EOs as an ingredient in virucidal disinfectants to
decrease alcohol content, and mitigate its harmful effects. In addition, the variation in
sensitivity of DENV-2 and CHIKV to the test EO-OH solutions suggests that these surrogate
viruses can be incorporated into the disinfection test against enveloped viruses to control for
variation related to differences in the viral envelope structure. We recognize that our work
has limitations, but it provided a first step in defining the potential utility of EOs as
disinfectants to control transmission of pandemic viruses transmitted via respiratory
secretions that contaminate surfaces. We have improved the discussion section of our
manuscript by including the above information.
Comment 2:
“The authors experimentally demonstrated the virucidal activity of an EO blend with
DENV-2 and CHIKV. However, the contribution of each EO component to the virucidal
activities by binding to viral envelope protein is only a speculation based on in silico
modeling. These EO components are commercially available. Including the experimental
results that support their hypothesis by using any of these EO components would
significantly increase the value of the findings of this manuscript”
We agree with the reviewer that evaluation of the antiviral effects of the EO components
would increase the value of our manuscript. This analysis should be included in a future study
to determine not only the contribution of each of the 42 compounds present in the EO blend,
but also to investigate the additive, synergistic and antagonistic effects of the EO components
on their virucidal action.
In silico and in vitro evidence suggests that sesquiterpene
2
hydrocarbons and oxygenated monoterpenes in specific ratios may account for the antiviral
action of the EOs (references 11-13). Recently, we have documented a variation in antiDENV effect related to variation in oxygenated monoterpene content (reference 21). We have
also documented (reference 31) a better
in vitro anti-DENV effect of trans-β-caryophyllene
and geranyl acetate compared to
p-cymene, limonene and neral, all of which are
terpenes identified in the test EO blend. Although our docking analysis results did not
confirm the role of terpenes in the virucidal action of the test EO blend, they have
supported
in vitro results suggesting the antiviral action of sesquiterpenes
and oxygenated monoterpenes against enveloped viruses. We have improved the discussion
section of our manuscript by including the above.
Comment 3:
“If the virucidal effect of EO is based on the specific interactions between the EO
components and viral envelope proteins I have following questions:”
1) “Is the virucidal activity of EO universal to many envelope viruses like alcohol,
considering the variety of envelope protein(s) that is unique to each virus?”
The virucidal activity of EOs can be universally applicable to viruses that are enveloped. In
vitro
and in vivo studies (references 11-13) demonstrate similar mechanisms of antiviral
action. (i) EO terpenes inhibit a variety of glycoproteins found in enveloped viruses, which
are crucial to the virus entry into host cells. (iii). Being lipophilic, EOs can penetrate the viral
envelope and disintegrate the membrane. (iii) Os can cause viral expansion, which interferes
with the attachment process by which the virus enters host cells. In addition, EO components
can inhibit lipid metabolism pathways in the host cell, which are crucial for the assembly of
new enveloped virions. The discussion section of our manuscript has been improved by
including the above information.
2) “If viral proteins, not the lipid envelope, are the target molecules of EO components
for their virucidal activity why the authors focused on only envelope viruses?”
As aforementioned, EOs can act on enveloped viruses through a variety of mechanisms that
do not target envelope glycoproteins alone. Our study focused on enveloped viruses because
studies reveal that enveloped viruses tend to infect more host species and are more likely to
be pandemic than non-enveloped viruses (reference 25). Coronavirus, influenza viruses,
Ebola virus, and HIV are the most important pandemic viruses, all of which being
RNA enveloped viruses.
Comment 4:
I sense some confusion in the arguments of the authors. If the authors believe the
interactions between the EO components and specific viral proteins, not the envelopes, are
the mechanism behind the virucidal activity of EO why the authors can argue EOs can be
non-specific disinfectant such as alcohol

We have argued that the test 12% EO + 10% OH solution was highly effective in inactivating
DENV-2 and CHIKV, which could be extended to other enveloped viruses. This is because:

3
(i) viral envelopes have common features (references 29, 33); (ii) the test solution was
effective against both viruses despite differences in the structure of their envelope proteins;
and as mentioned, (iii) EOs and alcohols can disrupt the viral envelope due to their lipophilic
nature, and (iii) EOs can cause viral expansion by blocking the attachment process. Our
results demonstrate that EO alone not only has disinfectant activity, but it also shows
synergistic activity with OH against two enveloped viruses. This synergistic activity may
involve all of the aforementioned mechanisms of action of EOs. We have improved the
discussion section of our manuscript by including the above.
Comment 5. Line 30
“influenza virus H1N1” should be “influenza A virus.”
Thanks for the comment. We have corrected the virus´s name (line 30).
Comment 6:
“Table 2: Please clearly indicate the unit of numbers in the table. Are they survival rate
in percent?”
Thanks for the comment. We have added a file in Table 2: ..”..Dilution / percentage of
viability..”
Comment 7:
Table 6 and 7: They did not appear in the text. They must be removed if not needed”.
Table 6 appeared in the text of the manuscript version 1 (lines 158 – 159), and it appears in
the new version of the manuscript (lines 159 – 160).
We apologize for the error, the Table 7 did not appear in the text of the manuscript version 1.
We have included the following text in the new version of the manuscript ((lines 190-191):
…”Table 7 presents the EO compounds with the lowest binding energy with the CHIKV E1-
E2-E3 complex”.
Comment 8. Line 211.
“Virus envelope does not “mediate virus binding to the cell surface …”
We have rewritten the sentence as follows (lines 228-230): …“ The viral envelope plays an
important role in the membrane fusion process [29], envelope proteins are potential
extracellular drug targets with multiple strategies to inhibit entry of the virus into host cells
[30]…”
Comment 9. Line 213.
“…differ in the lipid content and…”: This statement requires reference”
We agree with the reviewer. We have included two references:
4
Reference 26 (line 213): Hitakarun, A.; Williamson, M.K; Yimpring, N.; Sornjai, W.; Wikan,
N.; Arthur, C.J.; Pompon J.; Davidson, A.D.; Smith, D.R. Cell type variability in the
incorporation of lipids in the Dengue virus virion.
Viruses. 2022, 14, 1-17.
Reference 27 (line 213). Sousa, I.P.Jr.; Carvalho, C.A.M.; Gomes, A.M.O. Current
understanding of the role of cholesterol in the life cycle of alphaviruses.
Viruses. 2020, 13,
1-12.
Comment 10. Line296.
“ Did the authors mix several EOs in various proportions for the experiments? Does this
mean the authors used various EO blends, not a single mixture, during the study? The
authors need clearer definition of the EO blend (or blends) for the sake of
reproducibility”.
We used a single mixture of EOs distilled from seven Colombian aromatic plants. We have
rewritten the text in the Methods section for clarity (lines 273-275): ..“Pure EOs from seven
aromatic plants grown in Colombia were used to prepare an EO blend. Then, a stock solution
(6x10
6 µg/mL) of the EO blend was prepared in DMSO and used to prepare the test solutions
for analyses of the disinfectant activity (Table 1)..”.
Comment 11. Line 330.
“Did the authors use only water in the control while the EO blend was prepared in
DMSO?
We used only water in accordance with the German DVV/RKI guideline [reference 20].
Comment 12. Line 334.
“What does it mean “… with threefold serial dilutions of…on a logarithmic scale at base
10”? Did the authors make 3- fold dilutions or 10-fold dilutions?”
We apologize for the typing mistake. We made 10-fold dilutions (line 341-342).
3. Discussion
Cleaning virus-contaminated hands and surfaces is essential to
infection control and viral disease prevention [4]. OH
-based solutions
are utilized as disinfectants to control the transmission of human
pathogenic viruses. However, frequent and prolonged use of OHbased disinfectants may be harmful to health and the environment [7-
9]. EOs in the form of natural products are generally recognized as safe
(GRAS) by the FDA (Food and Drug Administration, U.S.A.), and their
use is permitted [23]. Many studies have explored using EOs as
potential antibacterial and antifungal alternatives to commercial
disinfectants [14,24]. In contrast, scientific evidence supporting the
potential of EOs as disinfectants against enveloped viruses is very
limited. Our study focused on enveloped viruses, studies show
that enveloped viruses tend to infect more host species and are more
likely to be pandemic than non-enveloped viruses [1, 25].

5
The present study evaluated the virucidal disinfectant activity of
solutions containing EO and OH against two surrogate viruses for
pathogenic enveloped viruses. The results show that a solution of 12%
EO combined with 10% OH reduced up to > 4.0 log10 TCID
50 (100%
reduction) of both viruses within 1 min of exposure. In addition,
solutions containing EO without OH also exhibited virucidal action
(100% reduction) against both viruses after 10 min (12% EO) and 30 min
(3% EO) of exposure. We did not observe a 100% reduction in either
virus with the 10% OH solution, but when combined with 12% EO, a
strong virucidal activity was observed. It appears that low
concentrations of EO and OH are insufficient to inactivate human
pathogenic enveloped viruses. Romeo et al. [18] did not observe
virucidal activity of a formulation containing 3.3% EO (
Melaleuca
alternifolia
) combined with 5.3% ethanol against the coronavirus HCovOC43 after 30 min of exposure.
To evaluate the virucidal disinfectant activity, we have used two
enveloped viruses, which differ in the lipid content [26,27] and protein
structure [20,22] that comprise the viral envelope. The results indicated
that DENV-2 was more sensitive to the action of test solutions than
CHIKV. We hypothesized that differences in the viral envelope
structure and its hydrophobic/hydrophilic nature might explain the
variation in sensitivity. The DENV-2 particle assembles and buds into
the endoplasmic reticulum of the infected cells where the envelope is
formed. The envelope has 90 head-to-tail dimers of the E protein
organized in a herringbone, with the M protein bound at the dimer
interface [28]. On the other hand, CHIKV assembles and budding
occurs at the cytoplasmic membrane, and the viral envelope comprises
the E1 and E2 glycoproteins and a peptide (E3) arranged in trimers to
make 80 E1/E2 spikes [23]. A recent study [18], showed differences in
the sensitivity of enveloped viruses (human and feline coronaviruses)
to treatment with a mixture of tea tree oil and ethanol.
Enveloped viruses enter host cells primarily via endocytosis
following attachment to a cellular receptor [2,25,29]. Upon attachment,
viruses are engulfed into endosomes where the low pH triggers
conformational changes of the envelope proteins to drive fusion of the
viral envelope and endosomal membrane. The viral envelope plays an
important role in the membrane fusion process [29], envelope proteins
are potential extracellular drug targets with multiple strategies to
inhibit entry of the virus into host cells [30]. Studies suggest that EOs
could cause morphological alteration of the viral particle by destroying
the viral envelope through interactions between their terpene
constituents and viral proteins [11,13].
In silico and in vitro evidence
suggest that sesquiterpene hydrocarbons and oxygenated
monoterpenes in specific ratios may account for the antiviral action of
the EOs [11-13]. Recently, we have documented a variation in the antiDENV effect related to variation in oxygenated monoterpene content
[21]. We have also documented [31] a better
in vitro anti-DENV effect
of
trans-β-caryophyllene and geranyl acetate compared to p-cymene,
limonene and neral, all of which were identified in the test EO blend.

6
We performed a primary docking analysis to describe the
interactions between the 42 compounds of the EO blend and the
envelope proteins of DENV-2 (E) and CHIKV (E1-E2-E3). As in a
previous study [21], in the present study, we again found sesquiterpene
hydrocarbons and oxygenated monoterpenes showing good binding
affinities (-6.7 to -8.6 kcal/mol) with DENV-2 E protein. These terpenes
were accommodated in the βOG pocket and molecules that dock this
pocket can block the conformational change of the E protein required
for the fusion process [32]. As for CHIKV, seventeen EO compounds
docked the E1-E2-E3 glycoprotein complex, some bound to the E2
protein in a pocket of the β-ribbon connector peptides, which play a
role during virus entry helping to trigger E1 conformational changes
during the fusion process [33]. Other EO compounds bound to the E1
protein of CHIKV near the hydrophobic fusion loop, which mediates
membrane fusion [33]. According to the docking score values, EO
compounds exhibited better binding affinities against DENV-2 than
against CHIKV, which could partly explain the differences in
sensitivity to the test solutions revealed in the virucidal disinfectant
assays.
Little is known about the specific mechanism of action of EOs
against enveloped viruses. Mechanisms other than alterations of the
envelope protein structure have been proposed [11-13]. Being
lipophilic, EOs can penetrate the viral envelope and cause membrane
disintegration; they can cause viral expansion, which interferes with
the attachment process by which viruses gain entry into host cells; and
EO components can inhibit host lipid metabolism pathways, which are
crucial to ensure the availability of lipids to complete the assembly of
new enveloped virions. On the other hand, OH cause protein
denaturation and disruption of the viral envelope [5]. Ethanol (95%,
v/v) has broader and stronger virucidal activity than propanols (75%
v/v); isoproponol, due to its lipophilic nature, interacts favorably with
viral envelopes; and glycerol (80% v/v) and glycerol derivatives have
been described as virucidal agents against enveloped viruses [34,35].
We hypothesize that the EO and OH mechanisms mentioned here
could be involved in the strong virucidal disinfectant activity of the
12%EO +10%OH solution.
Results of this study demonstrate that EO alone not only has
disinfectant activity, but also shows synergistic activity with OH
against two enveloped viruses. This synergistic activity may involve all
of the aforementioned mechanisms of action of EOs. Further analysis is
needed to investigate the contribution of each EO compound and their
additive, synergistic or antagonistic effects on the disinfectant action of
a pure EO.

Round 2

Reviewer 1 Report

The authors revised the manuscript according to the recommendations. So I wish them acceptance.